# What does an Adversarial Color look like?

## Abstract

The short-answer: it depends! The long-answer is that this dependence is modulated
by several factors including the architecture, dataset, optimizer and initialization.
In general, this modulation is likely due to the fact that artificial perceptual systems
are best suited for tasks that are aligned with their level of compositionality, so
when these perceptual systems are optimized to perform a global task such as
average color estimation instead of object recognition (which is compositional),
different representations emerge in the optimized networks. In this paper, we first
assess the novelty of our experiment and define what an adversarial example is in
the context of the color estimation task. We then run controlled experiments in
which we vary 4 variables in a highly controlled way pertaining neural network
hyper-parameters such as: 1) the architecture, 2) the optimizer, 3) the dataset, and 4)
the weight initializations. Generally, we find that a fully connected network's attack
vector is more sparse than a compositional CNN's, although the SGD optimizer
will modulate the attack vector to be less sparse regardless of the architecture. We
also discover that the attack vector of a CNN is more consistent across varying
datasets and confirm that the CNN is more robust to attacks of adversarial color.
Altogether, this paper presents a first computational exploration of the qualitative
assessment of the adversarial perception of color in simple neural network models,
re-emphasizing that studies in adversarial robustness and vulnerability should
extend beyond object recognition.

## 1   Introduction

Recent works in *"NeuroAI"* have shown the importance of task optimization for the construction of
robust neural networks models that try to find a perceptual alignment between biological and artificial
neural representations (Dwivedi & Roig, 2019; Wang et al., 2019; Schrimpf et al., 2020; Conwell
et al., 2021, 2022; Doerig et al., 2022). In these works, authors often test a battery of neural network
architectures or optimization constraints to evaluate how well such models align with human visual
perception. Conversely, modern research in adversarial images has focused on creating a plethora of
adversarial attacks & defenses for modern machine vision systems when networks are *exclusively*
optimized via a cross-entropy loss to encode a compositional task such as object recognition.

In this paper we shift gears and will focus on optimizing networks to perform average color estimation,
where we will mainly *not* be evaluating the robustness success via performance curves, but rather
qualitatively assessing how these differences look like when neural networks are optimized to estimate
the average color of an image across a variety of training and testing conditions (Emery & Webster,
2019; Shamsabadi et al., 2020; Kantipudi et al., 2020). We are primarily motivated by this framework,
because we would like to take a step back in adversarial image research to investigate how such
qualitative and quantitative differences are modulated when the object recognition task is completely
removed from the picture, and neural networks are optimized to estimate the average color of an
image instead. Perhaps convolutional neural networks become more robust? Perhaps they will not.

Submitted to 4th Workshop on Shared Visual Representations in Human and Machine Visual Intelligence
(SVRHM) at NeurIPS 2022. Do not distribute.

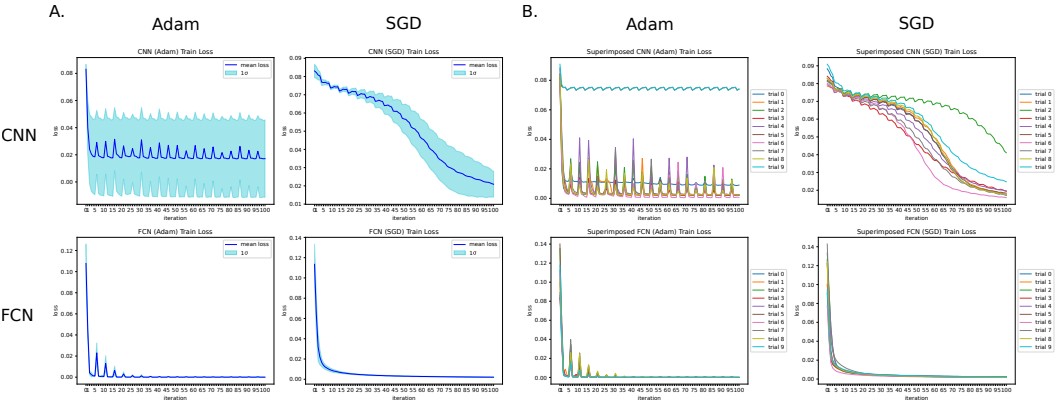

Figure 1: Visualization of the convergence of the loss function (MSE) over 20 separately trained Convolutional Neural Networks (CNN) and Fully Connected Networks (FCN) optimized with either Adam (10) or SGD (10) – totalling *40* trained neural networks over which we conduct our experiments.

Perhaps this will depend on the neural network architecture, dataset, optimizer and initialization or a combination of all these factors. This leads us to another question: What does an adversarial color look like in machines?

## 2 Training classic Neural Networks to compute Color Estimation

To investigate this question we will use two equi-parametric neural network families that have different approximational power induced by the computations pertaining to their architecture (See Appendix A). These are families of two *very basic* neural network types: Fully Connected Networks *(hereto FCN)* and Convolutional Neural Networks *(hereto CNN)* that are parameterized as seen in Table 1. The motivation for using these two types of architectures can be shown in Deza et al. (2020), that showed that while both CNNs and FCNs can approximate the average color of an image, FCNs easily arrive to a lower loss, given the correspondence of their architecture with the closed form expression of average color estimation $(C)$, which can trivially be expressed as the average luminance values per channel in an image:

$$C = \frac{\sum_i^N (I_i)}{N} \; ; \; \forall i \text{ pixels of image } I \tag{1}$$

In our experiments, both FCNs and CNNs were trained on the CIFAR-10 dataset with a Mean Square Error (MSE) loss and the last layer set to be a $3 \times 1$ vector, encoding the average RGB values of the image. In addition, each neural network was either optimized with SGD or Adam. The convergence of the training loss across all networks used in our experiments can be found in Figure 1 – as each FCN and CNN was optimized 10 times per each optimization procedure (SGD or Adam), totalling 40 neural network models.

Table 1: Training & parameter details of CNNs & FCNs

|  | CNN | | FCN | |
| --- | --- | --- | --- | --- |
|  | **Adam** | **SGD** | **Adam** | **SGD** |
| **# Params** | 61411 | 61411 | 61523 | 61523 |
| **# Epochs trained** | 50 | 50 | 50 | 50 |
| **# Trials** | 10 | 10 | 10 | 10 |
| **Learning Rate** | 0.01 | 0.001 | 0.0025 | 0.001 |
| **Lowest Loss** | 6.77e-4 | 1.58e-2 | 6.07e-7 | 1.75e-3 |

Additional details pertaining to attacking the neural networks and representations of color can be seen in Appendix B.

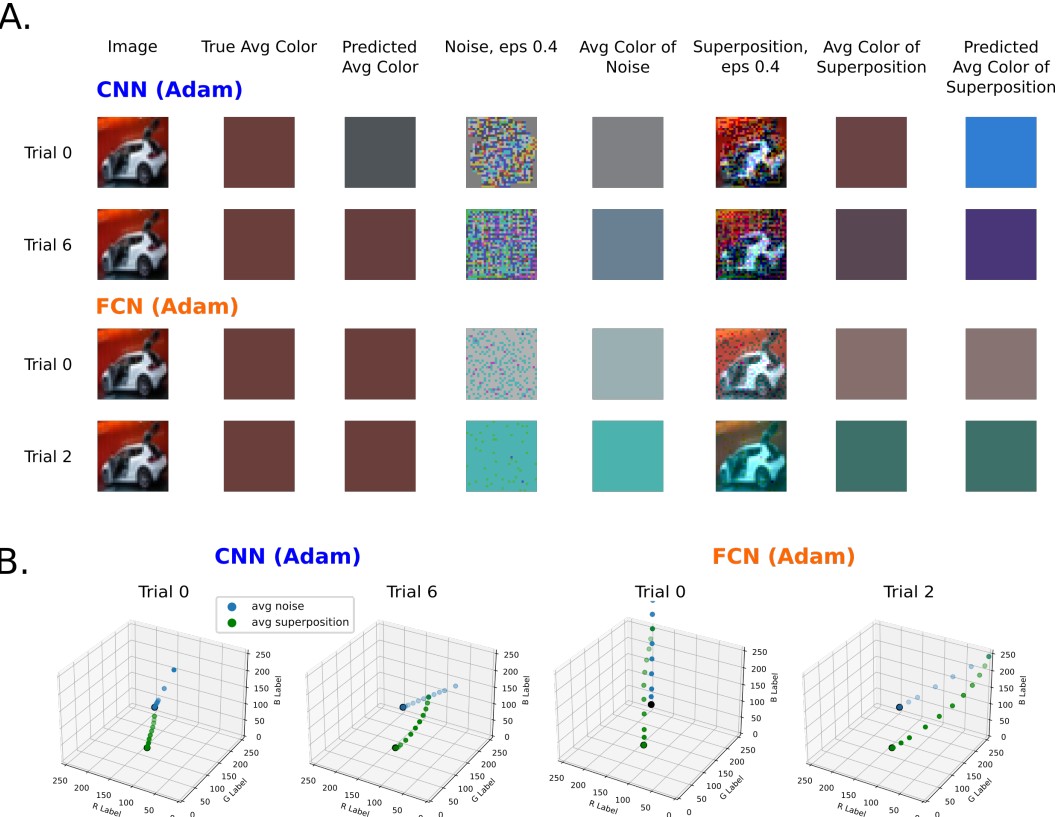

Figure 2: A. A diagram showing the differences and similarities of color-based adversarial attacks on the *same image* of two neural network trials across different architectures (CNN and FCN). B. The same image and their attacks in RGB color space demonstrating how each network is *"fooled"*. Notice that the average color of the superposition is curved by the average noise due to clipping (See Appendix B). All epsilon values and resulting attacked images are shown in the 3D plots (contra to inset A. that shows only a slide at $\epsilon = 0.4$)

## 3    Evaluation of Color Adversarial Attacks on Neural Network models

Thus, to shed light on the question of what an adversarial color look like, we must realize that this will be a distinctly different problem than one of the object recognition task, since there are no discrete classes by which we can determine when our network is fooled. Recall that the Fast Gradient Sign Method (FGSM) Attack can be formalized for the new adversarial image $\hat{x}$ by Goodfellow et al. (2014):

$$\hat{x} \leftarrow x + \epsilon \text{sign}(\nabla_x(\mathcal{L}(x, t, \theta))) \tag{2}$$

Following this paradigm for the color estimation task, we define an adversarial example as one where the adversarial noise *(a.k.a attack)* – the gradient of the loss w.r.t the input image – is bounded by epsilon in the set $\epsilon = [0.0, 0.0005, 0.001, 0.005, 0.01, 0.1, 0.2, 0.4, 0.6, 0.8, 1.0, 1.2, 1.4, 1.6, 2, 5, 10]$ *maximizing* the loss of the average color estimation. Hence, 'fooling the network' takes a trivial meaning in the average color estimation task as it is linked to a *regression* problem (estimates of RGB), rather than a *classification* problem (one-hot vector encoding as done in object recognition). Of course, it is thus trivial to say "the machine has confused the average RGB color of an image", so our interest is knowing *by how much* is it confused[1], and how the adversarial attack/vector/noise looks like.

As previously suggested in the introduction, the adversarial color is thus modulated by several factors including: 1) the architecture, 2) the optimizer, 3) the dataset, 4) the multi-trial analysis (i.e., different

---

[1]Notice that in the adversarial color landscape there is no notion of mis-classification accuracy, but rather difference in magnitude of MSE color estimation across color channels of the target and the prediction.

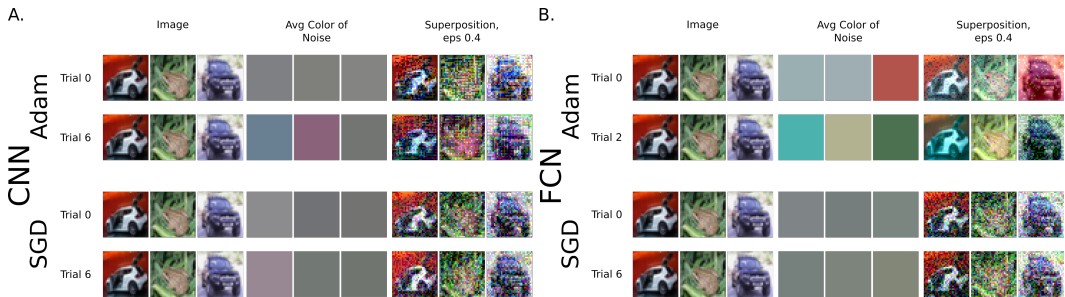

Figure 3: A small sample of the comparison of the effects of Adversarial Attacks on Average Color Estimation to both CNNs and FCNs when they are modulated by their optimizer. No clear differences emerge with the exception of an interaction between Adam + FCNs that reinforces a sparsity constraint on the Adversarial Attack. This can be observed in the Superposition, and also in Figures 6,7.

weight initializations of the same architecture). In the following subsections, we do our best to provide a holistic understanding of each one of these phenomena.

## 3.1 The Architecture and Optimizer: Interactions and Divergences

The effect of the optimizer does not affect our CNN models overall. The pattern of noise vectors is always dense and strangely off-set in the border with no noise (as if the CNN's stride were to implicitly direct the CNN to ignore the borders of the image: See Figures 6,7). Further investigation is required to explain this phenomena although some have found that this naturally emerges in CNNs trained for object recognition (Alsallakh et al., 2020; Yuan et al., 2021).

For the FCN, we see that Adam accentuates a sparse attack vector for many of the trials in our experiments. Whether this is a feature of a bug, requires more experiments, but this comes to a surprise as FCNs could opt for a theoretically stable weight vector that computes the exact average of all luminance channels through its layers (Figures 2,3).

## 3.2 Control Conditions: The Image Dataset & Individual Differences

We further examined the effects of varying the testing dataset for the color estimation task to challenge each neural network's robustness and learned perceptual representations. Rather than using MNIST (LeCun & Cortes, 2010), Places (Zhou et al., 2017), or ImageNet (Russakovsky et al., 2015), we first experiment with the simplest variations of color estimation from the CIFAR-10 test dataset: their vertically mirrored/inverted version, and a set of solid colors based on the original testing images. Notice that in all cases, the ground truth average color is *preserved*, but the image structure has been varied.

This variation is critical in testing for the color estimation task because we hypothesize that any rotational transformation should not affect the global structure of the color estimation attack for a FCN that does not have any explicit locality prior as a CNN does. Here, our initial expectations were that CNNs would also vertically flip their noise vectors (as these networks perhaps can not avoid trying to parse image identity even if it is not explicitly encoded in their loss function), while FCNs would stay invariant to such rotational shift.

We found that the attack vector of FCNs were not preserved during the vertical flipping (and in some cases also changed pattern; See Figures 6,7). CNNs, on the other hand, did seem to mildly preserve a structural bias when the image was flipped, but a further quantitative analysis is required to verify such a claim. More surprisingly, when we rendered solid colors that stemmed from such testing stimuli, CNNs failed to estimate color even with no adversarial attack, while FCNs did not struggle to accurately compute error. This would imply that CNNs are not learning an equipartite weighting scheme in their learned filters, as we would have initially expected[2].

---

[2]Although see Deza et al. (2020) that shows that CNN's first layer filters develop random weights to compute color, rather than Gabor-like structures, as CNNs learn to ignore edge cues to estimate color.

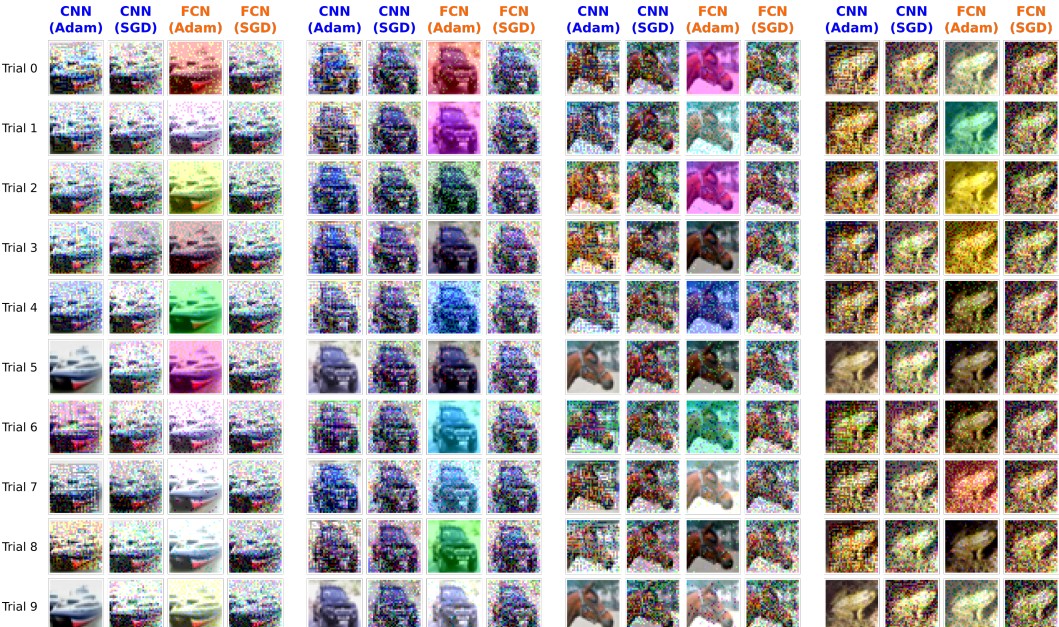

Figure 4: Individual differences in color perception from neural networks that are randomly initialized are shown. Such variations include architecture and optimizer across several images. By seeing these color-based adversarial attacks, it suggests that each neural network has learned to compute its "own" notion of color, analogous to what has been shown in humans (Emery & Webster, 2019).

Finally, extending our overall analysis to multiple trials, we see an interesting pattern of results as shown in Figure 4, where all neural network models seem to "learn to compute color" in a very different way (Lafer-Sousa et al., 2015; Emery & Webster, 2019) which can be visualized through the different attack structures in color space. What is even more puzzling is that on occasion, the final output of the adversarial attack will also be modulated by the input image. There are some cases where the attack is sparse, and other cases where it is dense (see Trial 2 for FCN optimized via Adam). This is an interesting interaction that we did not expect to find before running these experiments and requires further investigation which can not be covered in the scope of this paper.

## 4 Discussion

Circling back to the question that initially motivated this paper – *What does an Adversarial Color look like?* – we realize that the answer is still complex even if our focus in the paper pertains only to oversimplified models of machine vision. We have found that CNNs on the whole produce random noise vectors while fully connected models produce sparser noise vectors contingent on their optimization procedure. CNNs also chose to ignore the borders of the image to compute colors, unlike FCNs that would use the overall image information. Critically, FCNs seem to be better estimators of the average color of an image than CNNs even if CNNs are *more robust* to adversarial color attacks which is reminiscent of an accuracy-robustness tradeoff for object recognition-trained neural networks (Tsipras et al., 2018).

Indeed, perhaps a more accessible question for an extended version of this paper is: how do these attacks differ when compared quantitatively and qualitatively to neural networks trained to do object classification? And most importantly, do the adversarial attack patterns that arise across the different variations of neural networks optimized to do color estimation hold any resemblance to those that also fool a human observer? What are the effects of *adversarial training* for average color estimation? Future work will explore the perturbations performed on our machine vision models on humans, where we may be able to find that under the color estimation loss, such attacks fool humans the same way (Elsayed et al., 2018; Feather et al., 2019; Harrington & Deza, 2022; Feather et al., 2022).

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

## A    Neural Network Architecture Details

```
CNNet(
  (conv1): Conv2d(3, 6, kernel_size=(5, 5), stride=(1, 1))
  (pool): MaxPool2d(kernel_size=2, stride=2, padding=0, dilation=1, ceil_mode=False)
  (conv2): Conv2d(6, 16, kernel_size=(5, 5), stride=(1, 1))
  (fc1): Linear(in_features=400, out_features=120, bias=True)
  (fc2): Linear(in_features=120, out_features=84, bias=True)
  (fc3): Linear(in_features=84, out_features=3, bias=True)
)

FCNet(
  (fc1): Linear(in_features=3072, out_features=20, bias=True)
  (relu): ReLU()
  (fc2): Linear(in_features=20, out_features=3, bias=True)
)
```

## B    Implementational Details of Adversarial Attacks on Color Space

The implementation and visualization of the adversarial attacks for the color estimation task took place in two distinctly different color spaces. For the sake of quicker convergence in training, and smoother intervals of adversarial attacks, each CIFAR-10 input image was scaled down from integers in the space of [0, 255] so that its pixel values fit into the normalized range of [-1, 1] as flots.

Visualization of the adversarial attack took place in the un-normalized (original) range of [0,255] so that the images would have an appropriate meaning and rendering in the RGB color space. This transformation to [0,255] range only happened within the final stages of visualization so as to not interfere with the attack.

One consequence of visualizing images in any color space is that out-of-bounds values are not renderable/visualizable. In our context, this referred to values not in the range [-1,1] (e.g. -2.3, 4). Thus, whenever the neural networks models predicted average color of the adversarial image with values outside of the range [-1,1] – which happened quite often with $\epsilon = 0.4$ and higher – or when the adversarial image itself was created with too strong of a noise perturbation, we bounded these values to stay within [-1,1]. This is why the adversarial superposition is curved in figure 2.

This also affected our evaluation of the average loss of the networks on adversarial datasets as it capped the maximum distance between a label and prediction - bounding the regression problem. Thus, the MSE loss was never unreasonably high because the color space was bounded.

Future work will also explore how the color space may take into account the nature of the adversarial attack (RGB vs LAB or HSV).

## C    Supplementary Experiments

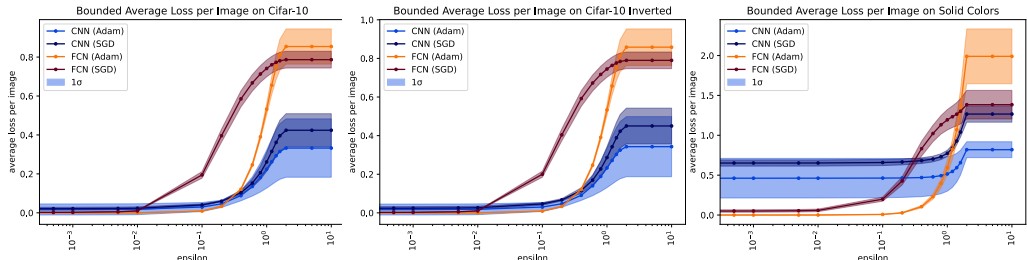

Figure 5: Visualizations of the bounded average loss of the 4 architecture types averaged over 10 trials, when tested on the adversarially attacked datasets with increasing epsilon. CNN's appear more robust than FCN's those approximation power of FCN's is evident in the solid color experiments. Individual trials can be see in Figure 8,9.

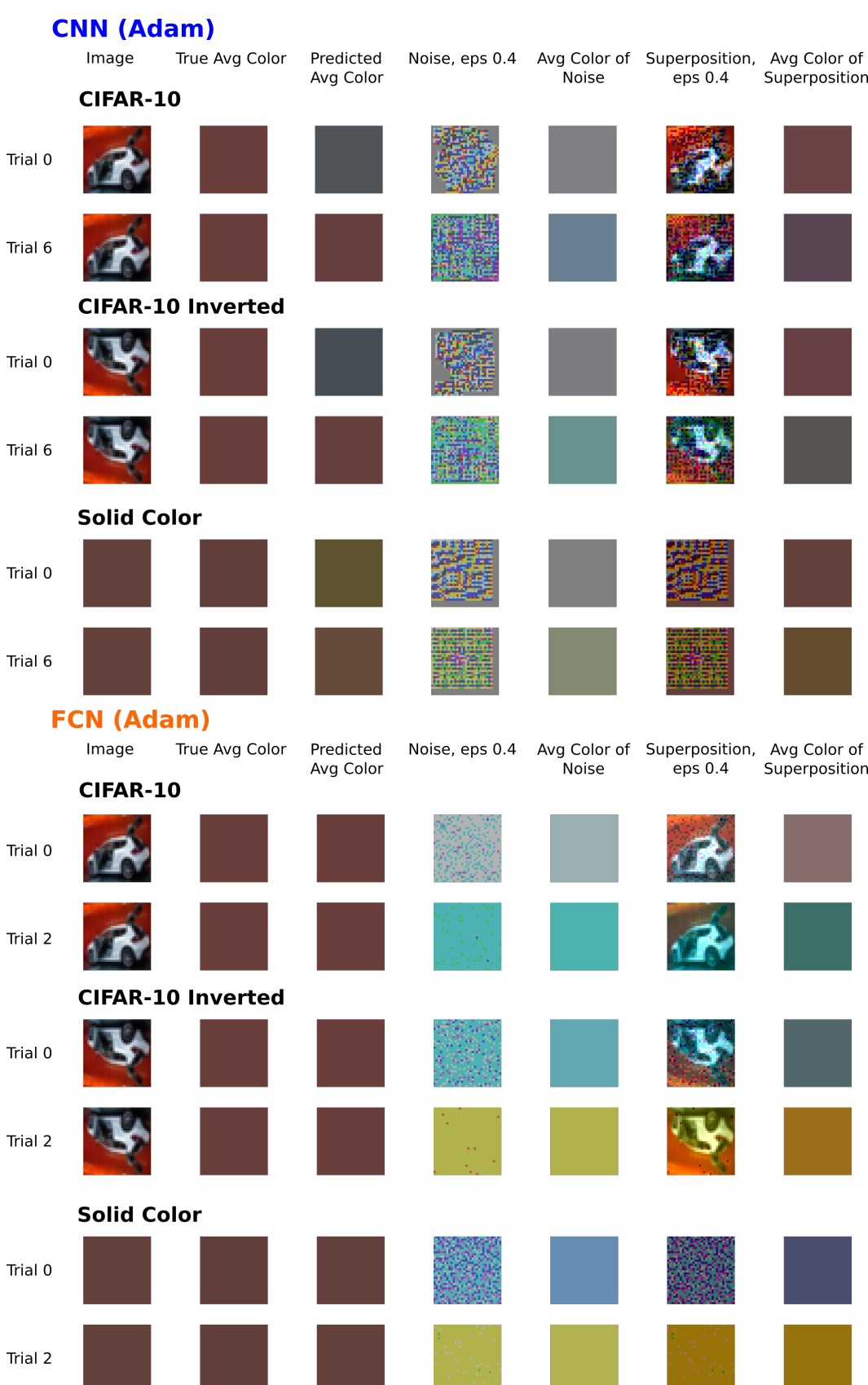

Figure 6: Full schematic visualizing the adversarial attack for Adam optimized neural networks, including ground truth, prediction and averages of different images (original, inverted and solid).

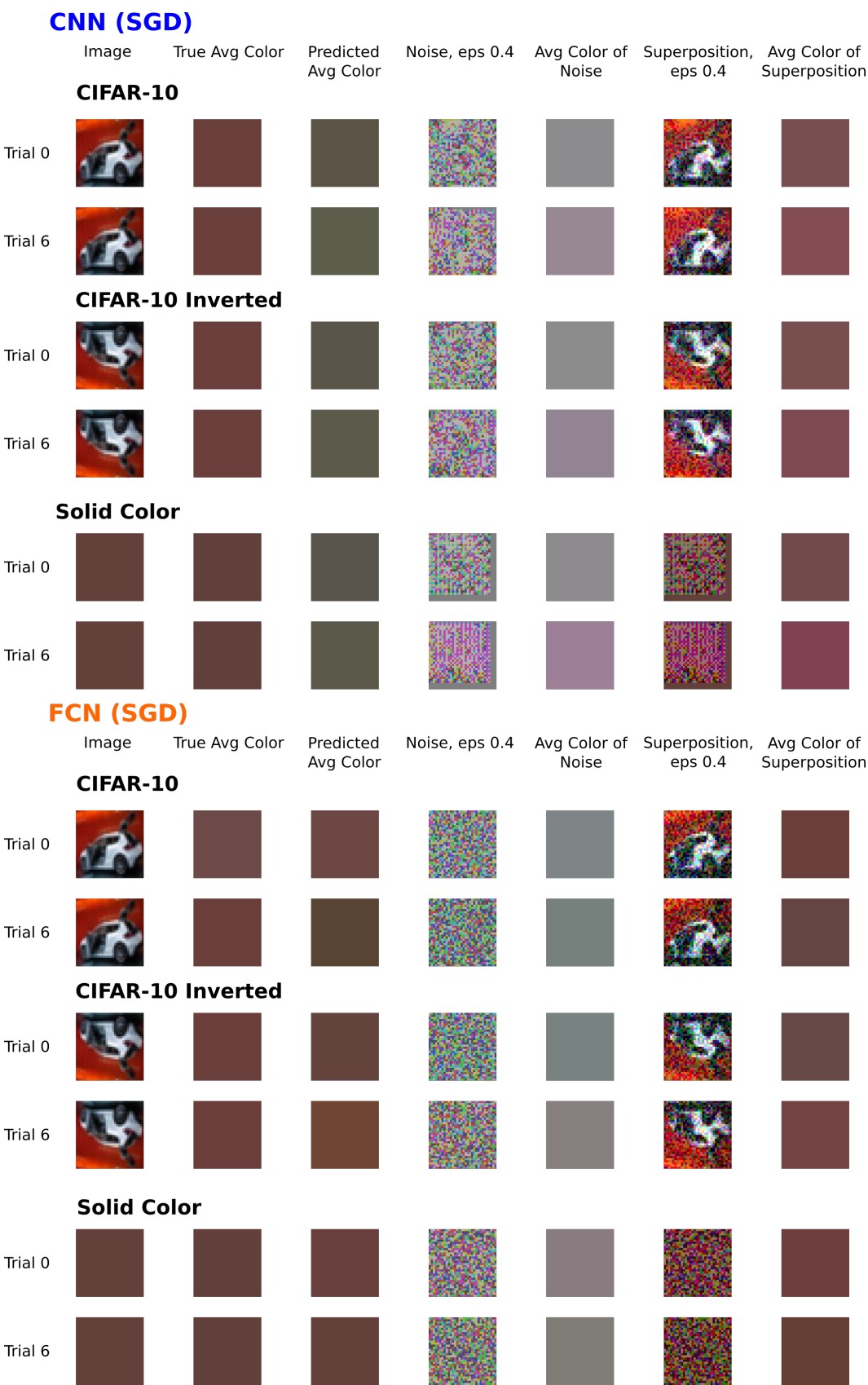

Figure 7: Full schematic visualizing the adversarial attack for SGD optimized neural networks, including ground truth, prediction and averages of different images (original, inverted and solid).

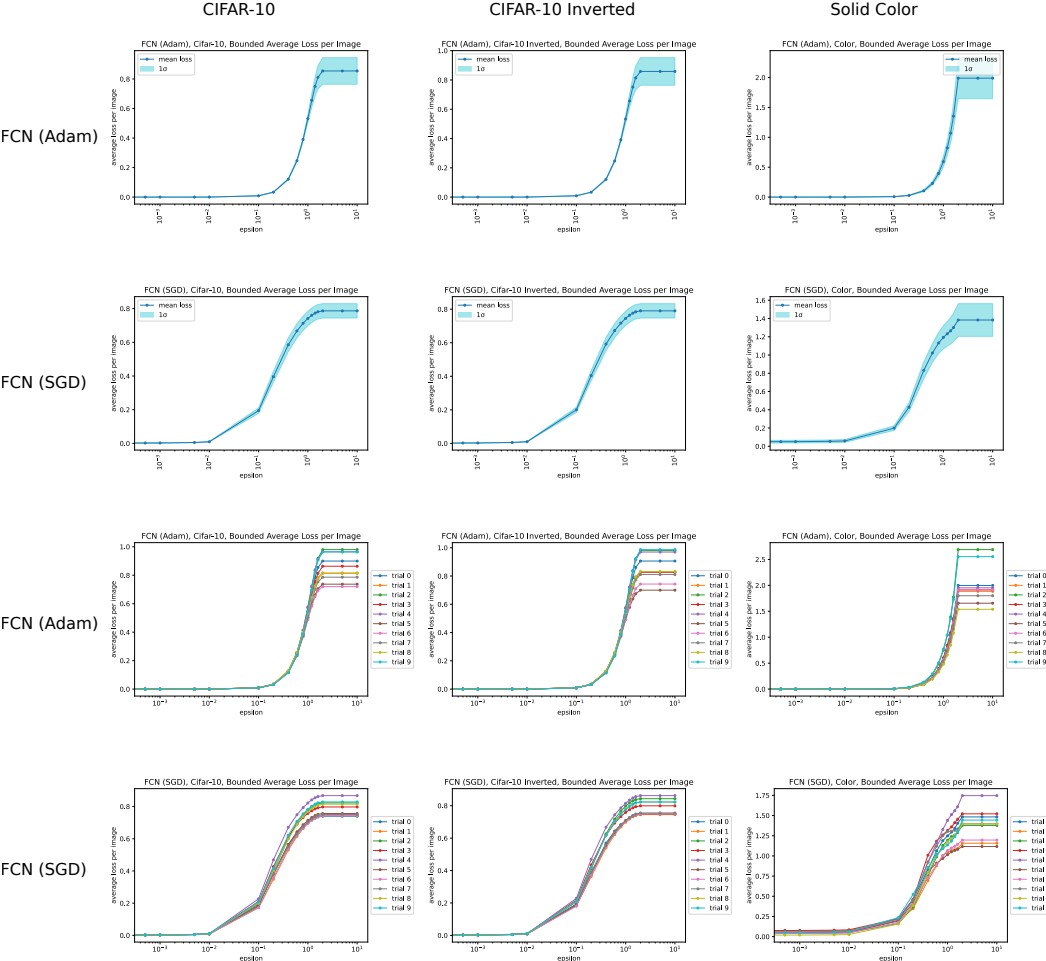

Figure 8: Quantitative estimation of the loss (computed via MSE) in normalized color space between the ground truth and the prediction for Fully Connected Neural Networks. Average Loss is visualized on the top, while the individual trial losses are visualized in the bottom. Notice that *all* FCNs compute color extremely well with near zero error.

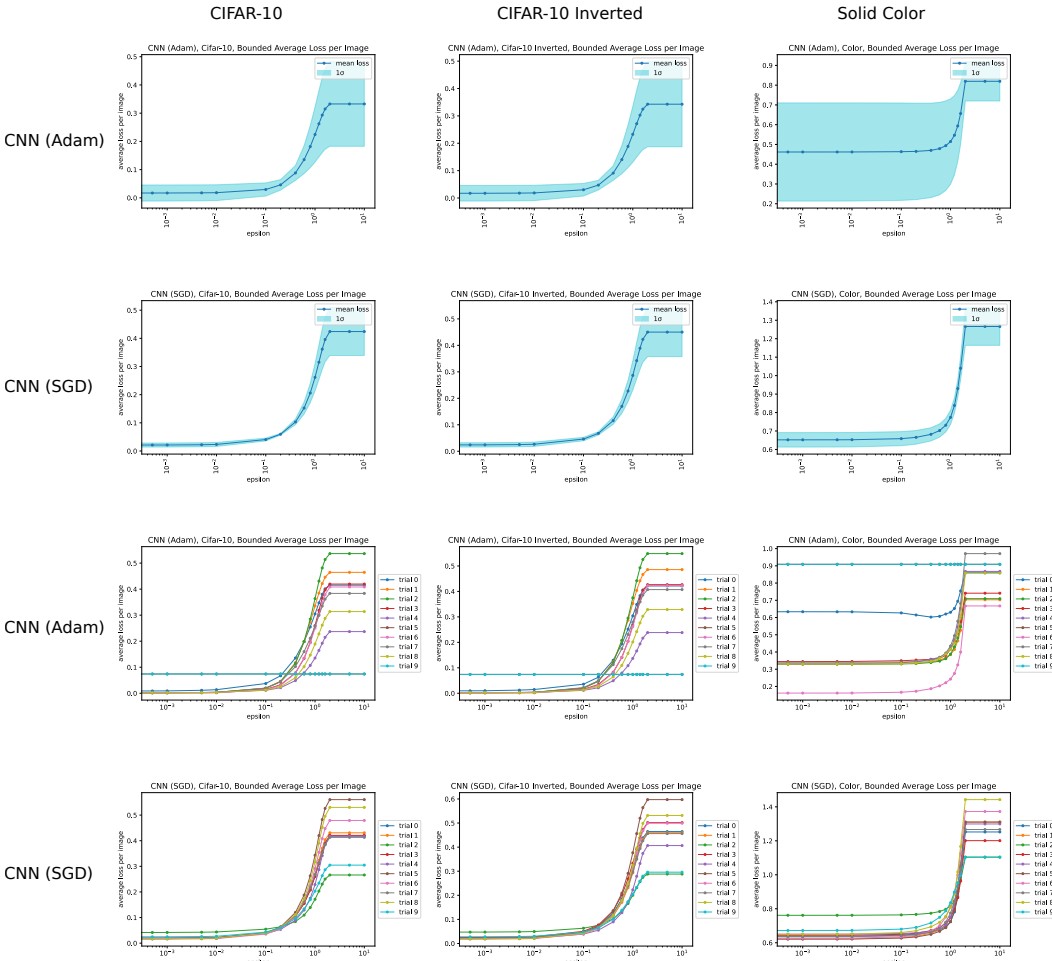

Figure 9: Quantitative estimation of the loss (computed via MSE) in normalized color space between the ground truth and the prediction for Convolutional Neural Networks. Average Loss is visualized on the top, while the individual trial losses are visualized in the bottom. Notice the variability of CNNs to compute average colors specially for solid colors.

