# OpenReview forum: "What does an Adversarial Color look like?"
_NeurIPS.cc/2022/Workshop/SVRHM — SVRHM Poster_

### Official Review · Reviewer_9viF · 2022-10-14

**Rating:** 4
**Confidence:** 4

**Review:**

This work aims to investigate foundamental properties of adversarial patterns of neural networks. For this, the authors focus on the very simple task of average color regression.

## Pros
- The authors construct a simple regression task to isolate the influence the network architecture and the regression problem itself has on the adversarial patterns found.
- This study investigates the influence of architecture and optimizer on the adversarial patterns found, which can be a valueable insight to better understand why adversarial exist and are introduced to neural networks.

## Cons
- The authors should set their approach and findings into context of the previous studies in the field of adversarial attacks
- I encourage the authors to investigate more attacks to make their findinings more general. The FGSM attack leveraged is an attack with limited power that also introduces certain biases into the adversarial patterns found.
- The description of the network architectures investigated is missing some details: Is it really correct that the CNN does not use any non-linear activation function (cf. Appendix A)?
- I have some doubts about the validity of the results: The authors say they use the FGSM attack (eq. 2) to generate adversarial patterns. Due to the sign operation, the resulting perturbation will be zero for some pixels (i.e., sparse), only if the gradient $\nabla_x\mathcal{L}$ is exactly zero. It would be quite surprising if the network really learned such a solution (which might happen due to the relu activation function) - thus, I'm wondering how it can be that the adversarial patterns are sparse (cf. L85)?
- The range of investigated $\epsilon$ values seems unreasonable: As the images are assumed to be in a $[-1, 1]$ range (cf. L261) a pixel can at most be changed by 2 before the value is clipped. As the attack either does not change a pixel at all (if the gradient is exactly zero) or it changes it by $\pm \epsilon$, using an $\epsilon$ value larger than 2 doesn't seem to make much sense.
- Section 3.1: It would have been interesting to apply this analysis to a single linear layer (which can achieve perfect regression error)
- Figure 1: Due to the small text size this figure is very hard to read.

## Conclusion
Weighting the pros and cons, I came to the conclusion that this work is not yet ready to be presented at a workshop. The authors list reasonable and interesting next steps in their conclusion that I very much encourage them to take to improve this paper!

---

### Official Review · Reviewer_8tJU · 2022-10-14
**What does an Adversarial COlor look like?**

**Rating:** 9
**Confidence:** 4

**Review:**

- The paper is well written, very clear and easy to follow
- The research question is clear, and the paper follows a well-defined scientific approach.
- The numerical evaluation is clear and consistent with what is being discussed.
- The supplemental material "supports" the paper adequately. being the main manuscript self-contained and enough to understand the work.
- I don't have any notes on language and/or style
- I would like to see how different colour spaces affect the results. But this has also been mentioned by the authors as future work.
- I would like to see the impact of different architecture types, optimizers and so on. But I understand this goes beyond the scope of this paper.
- Interesting topic and with the possibility to follow up.

---

### Official Review · Reviewer_wwtC · 2022-10-16
**Evaluation**

**Rating:** 5
**Confidence:** 2

**Review:**

This work studies the adversarial attack problem with FCN and CNN trained to perform whole-image color average regression. If I don't misunderstand it significantly, the idea is: whole-image color average regression is an almost trivial function to approximate since the perfect function is literally an average of all the pixels. I think the intuitive idea is that using these over-parameterized function approximators to approximate the averaging function will lead to some overfitting issues. By leveraging the adversarial attack, maybe we can understand more about how these overfitted functions overfit. Notably, this process of studying such a simplified function approximation problem may help us understand and disentangle the intertwined roles of the neural architecture, dataset, optimizer, initialization, etc. If my general understanding is somewhat correct, then this is an interesting problem if a clear conclusion can be drawn from it.

However, I highly recommend the authors improve the presentation as the conclusion is not yet clear enough. Some questions arise during my reading of this manuscript:

1. What can we learn from this problem? Or what is the major message?
2. What should be expected or surprising given the empirical results?
3. Do we have any theoretical insight? As color is quite redundant globally, intuitively, approximating this function might only need some sparse sampling to learn a relatively good approximation. Maybe it needs to reduce the error to really low to leverage all of the pixels densely. Is there any way to probe the behavior according to theoretical insights like this?
4. An averaging function of the color is so simple, why do the lowest losses shown in Table 1 have such a significant fluctuation? Is that due to bad tuning or a suboptimal learning schedule or so?

I have many questions like the above. It is like I have misunderstood here and there. So I highly suggest the authors make an effort to polish the presentation and formulate the problem clearly and concisely.  For a simplified problem like this, it will be helpful to give the readers a crystal clear conclusion rather than something like "it is still complex." Otherwise, the power of simplification is gone. I might be asking a lot out of a workshop paper. But I really hope to learn something more from this work. If the authors can make some significant effort to polish the paper and promise to deliver a clear message. Then I would be happy to accept it. So my rating is between 5 and 6, leaning towards five given its current presentation.